# Workshop version: How hard are computer vision datasets? Calibrating dataset difficulty to viewing time

**David Mayo**[*]
CSAIL & CBMM
MIT

**Jesse Cummings**[*]
CSAIL & CBMM
MIT

**Xinyu Lin**[*]
CSAIL & CBMM
MIT

**Dan Gutfreund**
MIT-IBM Watson AI
IBM

**Boris Katz**
CSAIL & CBMM
MIT

**Andrei Barbu**
CSAIL & CBMM
MIT

## Abstract

Humans outperform object recognizers despite the fact that models perform well on current datasets. Numerous efforts exist to make more challenging datasets by scaling up on the web, exploring distribution shift, or adding controls for biases. The difficulty of each image in each dataset is not independently evaluated, nor is the concept of dataset difficulty as a whole currently well defined. We develop a new dataset difficulty metric based on how long humans must view an image in order to classify a target object. Images whose objects can be recognized in 17ms are considered to be easier than those which require seconds of viewing time. Using 133,588 judgments on two major datasets, ImageNet and ObjectNet, we determine the distribution of image difficulties in those datasets, which we find varies wildly, but significantly undersamples hard images. Rather than hoping that distribution shift will lead to hard datasets, we should explicitly measure their difficulty. Analyzing model performance guided by image difficulty reveals that models tend to have lower performance and a larger generalization gap on harder images. We release a dataset of difficulty judgments as a complementary metric to raw performance and other behavioral/neural metrics. Such experiments with humans allow us to create a metric for progress in object recognition datasets. This metric can be used to both test the biological validity of models in a novel way, and develop tools to fill out the missing class of hard examples as datasets are being gathered.

## 1 Introduction

Numerous efforts exist to build better evaluations for object recognizers. Broadly, these fall into four categories. Those that probe distribution shift, like ImageNetV2 [1]. Those that add scale like OpenImages [2]. Those that explicitly attempt to make images more difficult for models by adversarially selecting them, like ImageNet-A [3] or adding artificial corruptions, like ImageNet-C [4]. And those that attempt to explicitly control for biases like ObjectNet [5]. These are responses to the fact that performance on standard benchmarks does not translate well to real-world conditions; 90% accuracy for one class in ImageNet does not mean 90% accuracy in the wild. In all four cases, these efforts have no objective guide, no metric that evaluates their progress towards generalization.

We measure an orthogonal quantity – how difficult images in these datasets are for humans. Distribution shift and bias control won't on their own address this problem if datasets are easy compared to what humans are capable of recognizing. And while scale helps, if datasets are heavily skewed

---

[*]Equal contribution. Website `https://objectnet.dev/flash` Corresponding author `dmayo2@mit.edu`

4th Workshop on Shared Visual Representations in Human and Machine Visual Intelligence (SVRHM) at the Neural Information Processing Systems (NeurIPS) conference 2022. New Orleans.

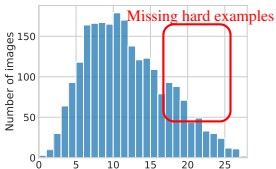
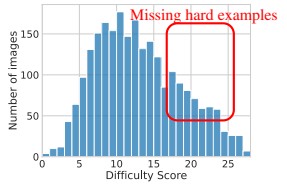

(a) ImageNet image difficulty distribution    (b) ObjectNet image difficulty distribution

Figure 1: The image difficulty distribution in ObjectNet and ImageNet. Difficulty here is defined as how many participants failed to recognize a given image across viewing times; easy images were almost always recognized even at short viewing times, while hard images were rarely recognized at short presentation times. Note that the difficulty of both datasets is roughly the same, and that hard images are under-sampled. Compared to what the human visual system can recognize, ImageNet and ObjectNet largely only test what can be recognized with short viewing times.

toward images that are easy for humans, the statistics of performance on such datasets may hide the real underlying performance trends of models on harder images.

Our contributions are: (1) a dataset of 133,588 human object recognition judgments as a function of viewing time for 4,771 images from ImageNet and ObjectNet, (2) the distribution of image difficulties for ImageNet and ObjectNet relative to what humans can recognize, shown in fig. 1, (3) a new analysis of model performance as a function of image difficulty, (4) a new metric for validating models' biological plausibility, predicting image difficulty, (5) a new subset of images from ObjectNet and ImageNet sorted by difficulty for use in neuroscientific and behavioral experiments.

## 2   Experiment

We performed an experiment with human subjects on Mechanical Turk and in the lab in order to determine the minimum amount of viewing time required before subjects could identify an object present in an image. See fig. 2 for the experiment overview. Additional procedures and experiment details can be found in the appendix.

We considered an image as recognized at some viewing time when half of the participants could classify it. Chance on the 1-out-of-50 task is 2%. Typical images by minimum presentation time are shown in fig. 6. Images that are quickly recognized by humans, easy images, are most similar to those seen in current datasets, while more difficult images include occlusion or difficult lighting.

An overview of accuracy as a function of viewing time online and in the lab is shown in fig. 3. Both experiments broadly agree with one another. In lab, the performance on short timings was significantly worse. We believe this is largely due to slow monitors which display the image for significantly longer than 17ms, roughly twice as long. At the high end, subjects in lab were nearly

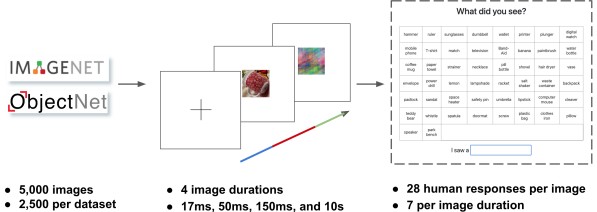

Figure 2: Overview of the experiment. (left) 50 images from 50 object classes were randomly selected from both ImageNet and ObjectNet to total 5,000 images; of which we analyze 4771. Images were cropped in a square around the object of interest and then shown to human subjects on Amazon Mechanical Turk and in a controlled laboratory setting. (middle) Participants first saw a fixation cross for 500ms, then the image for either 17ms, 50ms, 150ms, or 10s, followed by a mask. (right) After each image, subjects were given a 1-out-of-50 forced-choice task to identify the correct object class. Each image was seen by 28 subjects, seven for each of the four image durations. No subjects saw the same image twice.

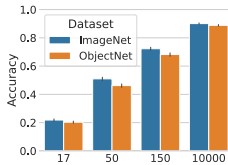 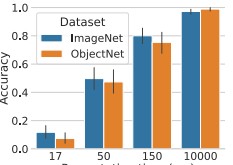 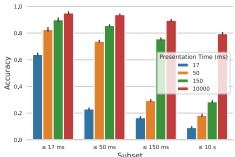

(a) Mechanical Turk experiments     (b) In-lab experiments     (c) Difficulty subset accuracy

Figure 3: Accuracy as a function of presentation time. The same images (5,000 online, 200 in lab) were presented at 4 timings. Results for both conditions were similar, although in-lab experiments achieved nearly 100% accuracy with 10 second viewing times, while halving the performance at 17ms compared to MTurk.

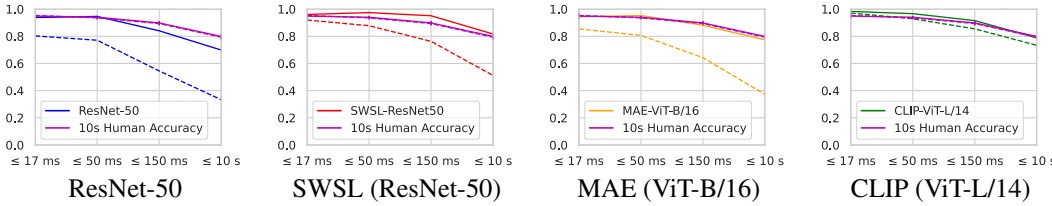

ResNet-50     SWSL (ResNet-50)     MAE (ViT-B/16)     CLIP (ViT-L/14)

Figure 4: Accuracy of four recent models on ImageNet (solid) and ObjectNet (dashed) as a function of image difficulty (extensive experiments with numerous additional models are available in the appendix). Model performance drops off significantly for the harder images. Harder images also show a much larger gap when testing on ObjectNet.

100% accurate, 10% higher than online. We believe this likely has to do with how distracted subjects were. In what follows, we focus on the Mechanical Turk experiments due to their scale.

**How hard are today's object recognition datasets?** We use minimum viewing time required for reliable recognition as a proxy for image difficulty. Datasets today are not gathered to control for difficulty and, indeed, when plotting the difficulty of images in ImageNet and ObjectNet, we find that the difficulty curve for these datasets is highly skewed; see fig. 1. Rather than plotting four bins, one for each viewing time, we plot a more fine-grained quantity: the total number of incorrect responses out of the 28 presentations of each image (7 participants at 4 timings). Images with few incorrect responses are easy: all participants at all timings could recognize them, even the short timings. Images with many incorrect responses are hard: few participants could recognize the images at only a few longer timings.

**What we can learn about models from hard images?** Machine accuracy varies as a function of the minimum viewing time required to recognize the object in an image. Most models, but not all, see a significant performance dropoff between the easy images and the hard images, see fig. 4. See the appendix for extensive results for dozens of models broken down by image difficulty. Note that this understates human performance as it shows the Mechanical Turk results. In-lab, even for the hardest images, humans have nearly perfect performance.

These results also show that the gap between ImageNet and ObjectNet performance increases as image difficulty increases. Likely, many phenomena such as distribution shift are much more acute for harder rather than easier images. Image difficulty can tease apart differences between recognition models that would otherwise be lost because of the skewed underlying difficulty distributions in current datasets. In fig. 5, we plot numerous object recognition models contrasting their performance on the easiest and hardest images. Rather than computing the absolute performance of models, that would naturally favor larger models with larger training sets, we measure the gap in performance between ImageNet and ObjectNet.

**Explaining image difficulty and testing how similar models are to humans.** If models are to not just perform well, but to also process images in ways that are similar to the human visual system, then they should contain a proxy for difficulty. We investigated three quantities computed from models that could be used to explain the difficulty judgments: c-score [6], prediction depth [7], and adversarial robustness [8]. See the appendix for plots of the correlation between these metrics and difficulty. This analysis revealed that images that require more viewing time for humans are harder for networks in several ways. They are learned much later in the training process, are predicted by

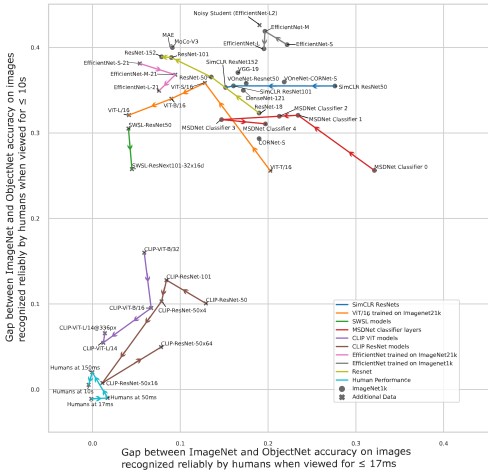

Figure 5: Model robustness trends as a function of image difficulty. The horizontal and vertical axes show the accuracy gap between ImageNet and ObjectNet for easy and hard images respectively. Each point is a model. Lines and arrows connect models that are part of the same family, pointing from small to large. Humans hover around zero, they are robust with respect to the distribution shift. As model size increases plain ResNets improve on easy images, but increase their gap on harder images. Extrapolating the performance of models shows that many current model families are likely to stop or radically slow down performance improvements on hard images, a less optimistic story than aggregate accuracy. CLIP stands out. Larger CLIP models tend toward only a small gap.

later model layers and require much smaller perturbation to successfully adversarially attack. This underscores that human minimum viewing time has many practical consequences for how networks process images. Logistic regression including all three metrics classifies images into the four difficulty subsets with 48.5% accuracy, far above chance, but leaving plenty of room to do more. New metrics for understanding how networks process images could be validated against this data.

## 3 Conclusion and related work

As a field, we have measured our progress by models' performance on tests created by splitting a random subset of images from large-scale web scraped datasets, labeled by a consensus among human annotators [9]. More recently, researchers have realized that there is significant value to testing models' abilities to recognize objects out of domain [1, 3, 4, 5].

Meding 2022 et al. [10] investigated differences in model performance by removing images found to be either "trivial" or "impossible for models". They also found humans could accurately classify images into these two categories. Geirhos 2018 et al. [11] studied human and machine performance under limited presentation time with test set images made more difficult by adding image corruptions.

Rather than focusing on scaling, distribution shift, or control for biases alone, we should also focus on dataset difficulty explicitly. Today's datasets skew toward being too easy by undersampling hard images. ObjectNet was designed for distribution shift and bias control and was not collected from the web, yet its distribution of image difficulties is remarkably similar to that of ImageNet. By focusing on ways to measure dataset difficulty as datasets are collected, we can better calibrate the entire community, and create the resources needed to push object recognition forward. In addition to just creating better datasets, understanding performance as a function of difficulty reveals radically different scaling curves for different models and approaches. It can also provide subsets of images that highlight different types of processing for neuroscientific or behavioral experiments.

While we performed extensive experiments to validate the approach presented here, measuring the difficulty of any one dataset is easy and cheap. Doing so with a model is free, but noisy, limited by how well the combination of c-score, prediction depth, and adversarial robustness explain the viewing time results. And of course, the validity of these metrics may go down with significant distribution shift. On the other hand, measuring dataset difficulty by sampling several hundred images and running an experiment on Mechanical Turk, only costs on the order of low hundreds of dollars per dataset and can be carried out quickly. The more critical the dataset, and the cost of object recognition failures, the more important doing so is.

## Acknowledgments and Disclosure of Funding

We would like to thank Ko Kar, Jim DiCarlo, Dan Yamins, and Martin Schrimpf for their helpful feedback and discussion about our experiments. We would also like to thank David Lu for his contributions to our early experiments and directions.

This work was supported by the Center for Brains, Minds and Machines, NSF STC award 1231216, the NSF award 2124052, the MIT CSAIL Systems that Learn Initiative, the MIT CSAIL Machine Learning Applications Initiative, the MIT-IBM Watson AI Lab, the CBMM-Siemens Graduate Fellowship, the DARPA Artificial Social Intelligence for Successful Teams (ASIST) program, the DARPA Knowledge Management at Scale and Speed (KMASS) program, the United States Air Force Research Laboratory and United States Air Force Artificial Intelligence Accelerator under Cooperative Agreement Number FA8750-19-2-1000, the Air Force Office of Scientific Research (AFOSR) under award number FA9550-21-1-0014, and the Office of Naval Research under award number N00014-20-1-2589, and award number N00014- 20-1-2643. The views and conclusions contained in this document are those of the authors and should not be interpreted as representing the official policies, either expressed or implied, of the U.S. Government. The U.S. Government is authorized to reproduce and distribute reprints for Government purposes notwithstanding any copyright notation herein.

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

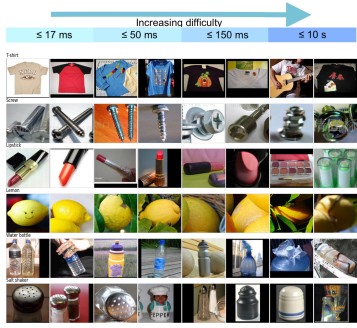

Figure 6: Images as a function of difficulty measured by minimum viewing time before recognition.

# A   Appendix

## A.1   Example images

## A.2   Dataset license

We release or presentation time judgments dataset under the Creative Commons Attribution 4.0 license. No personally identifiable data was collected from any subject.

## A.3   Image cropping procedure

1. We draw a bounding box around the object (we use existing bounding boxes for the ImageNet validation set and collect our own bounding boxes for ObjectNet from MTurk).
2. We initialize the cropping box to be the bounding box.
3. If the cropping box does not form a square, we extend the shorter side of the rectangular cropping box to form a square. If the image is not large enough to extend the shorter side of the cropping box, we pad it with black pixels to form a square.
4. We crop using the cropping box for the image. The cropped image will be a square.
5. We resize the cropped image to be 224x224 pixels.

## A.4   Mask generation

The masks were generated following the procedure used by [35]. Specifically, a Fourier transform was applied to each image to obtain the magnitude and phase components. Then, a random array with elements sampled uniformly from [0, 1] was added to the image phase component after which the magnitude and phase components were recombined via an inverse Fourier transform to produce the mask. Each image was paired with its particular phase-scrambled mask in the experiments.

## A.5   Experiment Procedure and Payment

Participants both in the lab and on Mechanical Turk were presented with a document informing them of the purpose, privacy, and risks associated with the experiment and soliciting their consent to participate (see fig. 7). Participants were then instructed as to how to carry out the experiment and were shown an example video as well as the list of image classes for their review before beginning. They were informed that they would not need to memorize the classes as the classes would be shown to them after each video. Participants were also encouraged to take breaks should they feel fatigued or otherwise uncomfortable. Example instructions are shown in fig. 8

After giving consent and reading the experiment overview. participants then completed two calibration steps for to estimate the size of their monitor and their distance from the screen for us to then size the videos appropriately to 8 degrees of visual angle. First, the participants are shown an image of a credit card and are asked to use a card of their own to adjust a slider to change the size of the card on the screen to the size of their card. Since credit cards are the same size around the world, this allows us to measure the pixel-to-inches ratio of the participant's monitor. Next, the participant completes a blind-spot test [33] that allows us to estimate the distance they are sitting from their screen. Together, these two measurements are sufficient to compute the desired video eccentricity. See fig. 9 for images of the calibration steps.

| | |
|---|---|
| Number of responses | 133,588 |
| Number of images | 4,771 |
| Number of presentaiton durations | 4 |
| Number of response per image | 28 |
| Number of ObjectNet images | 2,415 |
| Number of ImageNet images | 2,356 |
| Number of participants | 1,495 |

Table 1: Dataset statistics

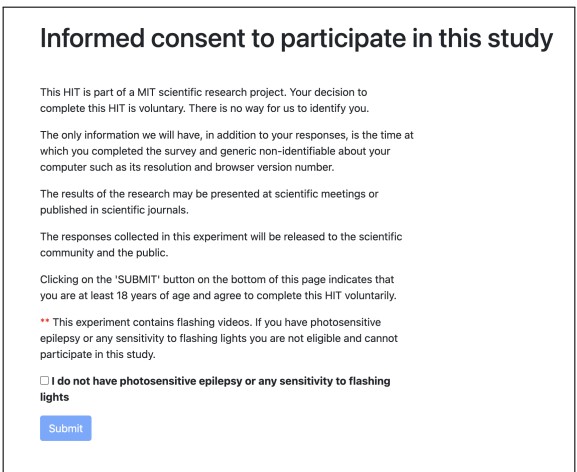

Figure 7: Informed consent page shown to participants before beginning the experiment.

The estimated hourly wage for participants on Mechanical Turk and in the lab was $10/hr and $20/hr respectively with approximately $15,000 spent in total on participant compensation.

## A.6 In-Lab Experiment Results

To corroborate our Amazon Mechanical Turk results, we selected 200 images shown to Turk workers to conduct the same experiment in a controlled laboratory setting. 12 individuals came to participate in the experiment in which they viewed and responded to all 200 images on our 144Hz refresh rate monitor with 1ms gray-to-gray time. After conducting the experiment, 3 individuals had seen each image at each of the 4 presentation times. When compared to the MTurk results for those same 200 images, the comparison is much as we would expect. The In-Lab accuracy with shortest image duration (17ms) is less than on MTurk which can likely be contributed to the use of our new, high refresh-rate monitor in the controlled environment. It is likely that MTurk workers' personal computers differ in their graphics presentation abilities which may result in the image being visible for slightly greater than 17ms on some monitors. On the other end, the in-lab experiments reported higher accuracy at the longest image duration (10s) which is also unsurprising as the in-lab participants completed the task in a controlled environment with no distractions and are likely more inclined to take the task seriously and stay focused. The results show no significant differences in accuracy at the intermediate image durations. See fig. 3 for a side-by-side comparison between MTurk and In-Lab results.

## A.7 Dataset statistics

We collected 28 human responses for each of 5,000 images (2,500 from ImageNet and 2,500 from ObjectNet). After reviewing response, 229 images were removed due to either being unrecognizable, mislabeled, or having been seen by the same worker twice despite safeguards in place to disallow it. Additional dataset statistics are listed in table 1.

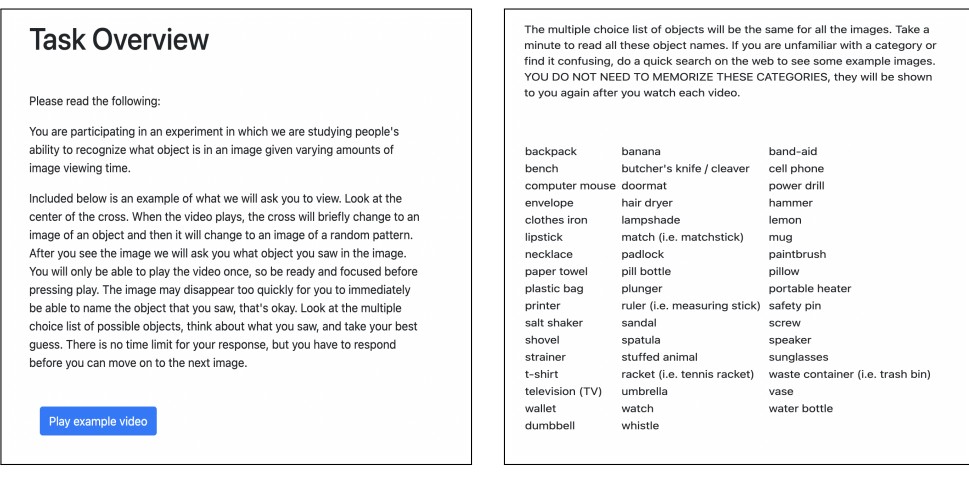

Figure 8: Instructions given to participants before beginning the experiment.

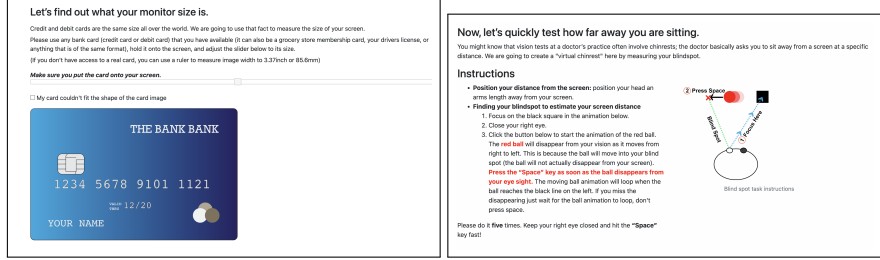

Figure 9: Images of the experiment calibration steps. The credit card task was used to measure the pixel-to-inches ratio of the subject's screen. The blind spot task provided an estimate of the subjects distance from their screen.

| Subset
Dataset | <= 17
Image-Net | Object-Net | <= 50
Image-Net | Object-Net | <= 150
Image-Net | Object-Net | <= 10000
Image-Net | Object-Net |
|---|---|---|---|---|---|---|---|---|
| ResNet-18 [13] | 92.7 | 73.7 | 89.8 | 63.0 | 76.2 | 41.7 | 55.7 | 23.3 |
| ResNet-18-80% | 87.6 | 66.4 | 85.0 | 60.9 | 71.2 | 38.3 | 51.6 | 20.8 |
| ResNet-18-60% | 84.8 | 67.2 | 85.7 | 57.4 | 70.2 | 38.1 | 51.1 | 20.8 |
| ResNet-18-40% | 83.1 | 58.4 | 83.1 | 52.0 | 63.5 | 36.5 | 48.7 | 18.8 |
| ResNet-18-20% | 78.7 | 46.7 | 74.1 | 44.5 | 60.7 | 26.0 | 38.3 | 13.3 |
| ResNet-50 [13] | 93.8 | 80.3 | 94.6 | 77.1 | 84.0 | 54.5 | 69.9 | 33.3 |
| ResNet-50-80% | 88.2 | 75.9 | 89.9 | 68.7 | 80.6 | 48.3 | 62.9 | 26.3 |
| ResNet-50-60% | 89.3 | 70.1 | 89.0 | 67.4 | 75.6 | 46.7 | 61.4 | 25.1 |
| ResNet-50-40% | 92.7 | 68.6 | 86.2 | 59.5 | 72.3 | 40.5 | 53.7 | 23.3 |
| ResNet-50-20% | 85.4 | 58.4 | 81.1 | 48.4 | 66.8 | 32.9 | 46.3 | 19.2 |
| ResNet-101 [13] | 94.4 | 85.4 | 94.3 | 78.8 | 88.6 | 59.6 | 74.7 | 35.9 |
| ResNet-101-80% | 92.1 | 74.5 | 92.5 | 72.9 | 81.9 | 51.7 | 65.5 | 30.0 |
| ResNet-101-60% | 91.0 | 75.9 | 88.3 | 67.1 | 78.8 | 46.7 | 61.7 | 27.6 |
| ResNet-101-40% | 89.3 | 67.9 | 87.9 | 60.0 | 74.8 | 40.2 | 56.9 | 22.2 |
| ResNet-101-20% | 85.4 | 51.8 | 83.1 | 50.5 | 66.6 | 34.4 | 51.3 | 17.1 |
| ResNet-152 [13] | 93.3 | 85.4 | 95.5 | 80.9 | 89.9 | 62.9 | 75.2 | 36.3 |
| ResNet-152-80% | 90.4 | 75.2 | 90.6 | 71.3 | 82.7 | 51.1 | 67.2 | 29.2 |
| ResNet-152-60% | 92.7 | 73.0 | 89.0 | 66.8 | 79.2 | 48.8 | 61.9 | 27.3 |
| ResNet-152-40% | 92.1 | 65.7 | 86.8 | 61.4 | 73.9 | 43.5 | 56.4 | 22.9 |
| ResNet-152-20% | 85.4 | 60.6 | 81.3 | 50.8 | 69.5 | 35.1 | 48.9 | 16.3 |
| CORNet-S [14] | 92.7 | 73.7 | 90.6 | 68.7 | 78.3 | 47.0 | 55.4 | 26.1 |
| VOneNet-Resnet50 [15] | 92.7 | 75.2 | 92.7 | 72.5 | 81.9 | 49.5 | 61.7 | 25.9 |
| VOneNet-CORNet-S | 90.4 | 68.6 | 90.0 | 63.2 | 78.8 | 44.1 | 58.3 | 22.4 |
| VGG-19 [16] | 91.0 | 74.5 | 88.6 | 65.7 | 78.2 | 45.9 | 60.0 | 22.9 |
| Noisy Student (EfficientNet-L2) [17] | 94.9 | 75.9 | 92.5 | 66.6 | 85.3 | 45.8 | 67.7 | 25.1 |
| DenseNet-121 [18] | 93.8 | 76.6 | 92.0 | 73.0 | 82.2 | 50.9 | 63.6 | 28.6 |
| MSDNet Classifier 0 [19] | 78.1 | 46.0 | 73.3 | 38.2 | 55.7 | 27.2 | 38.6 | 12.9 |
| MSDNet Classifier 1 | 87.6 | 64.2 | 84.6 | 54.6 | 70.0 | 37.1 | 51.1 | 19.0 |
| MSDNet Classifier 2 | 89.9 | 68.6 | 88.7 | 62.2 | 73.8 | 45.3 | 57.8 | 25.9 |
| MSDNet Classifier 3 | 89.9 | 75.2 | 88.9 | 66.6 | 73.3 | 45.8 | 57.8 | 26.3 |
| MSDNet Classifier 4 | 92.7 | 73.0 | 89.5 | 67.8 | 75.2 | 45.6 | 58.3 | 27.3 |
| SimCLR ResNet50 [20] | 88.2 | 60.6 | 84.5 | 59.3 | 70.0 | 46.1 | 56.9 | 21.4 |
| SimCLR ResNet101 | 92.7 | 76.6 | 88.3 | 68.8 | 80.0 | 52.9 | 64.3 | 28.8 |
| SimCLR ResNet152 | 93.3 | 78.1 | 90.2 | 70.0 | 81.4 | 52.7 | 67.5 | 32.2 |
| CLIP-ViT-B/32 [21] | 94.9 | 89.1 | 88.8 | 78.2 | 75.7 | 60.7 | 56.4 | 40.4 |
| CLIP-ViT-B/16 | 97.2 | 90.5 | 93.2 | 86.2 | 81.3 | 73.1 | 62.7 | 53.1 |
| CLIP-ViT-L/14 | 98.3 | 97.1 | 96.6 | 92.9 | 91.5 | 85.4 | 78.8 | 73.3 |
| CLIP-ViT-L/14@336px | 97.8 | 96.4 | 96.3 | 93.6 | 91.2 | 87.5 | 79.5 | 72.9 |
| CLIP-ResNet-50 | 91.0 | 78.1 | 81.6 | 70.1 | 65.0 | 55.0 | 42.4 | 32.4 |
| CLIP-ResNet-101 | 93.8 | 85.4 | 85.4 | 75.1 | 70.0 | 58.4 | 49.6 | 36.9 |
| CLIP-ResNet-50x4 | 93.3 | 85.4 | 86.4 | 77.5 | 73.6 | 64.7 | 51.1 | 40.8 |
| CLIP-ResNet-50x16 | 93.8 | 92.7 | 89.8 | 83.2 | 78.3 | 72.5 | 53.5 | 52.7 |
| CLIP-ResNet-50x64 | 98.3 | 90.5 | 94.5 | 89.6 | 86.0 | 82.6 | 65.5 | 60.6 |
| EfficientNet-S [22] | 89.3 | 67.2 | 92.0 | 63.8 | 78.0 | 44.3 | 63.9 | 23.5 |
| EfficientNet-M | 90.4 | 70.8 | 89.9 | 63.4 | 75.4 | 39.9 | 63.9 | 22.0 |
| EfficientNet-L | 93.3 | 73.7 | 92.8 | 68.2 | 83.9 | 47.4 | 68.7 | 28.8 |
| EfficientNet-S-21 | 96.6 | 91.2 | 95.2 | 80.7 | 89.1 | 63.7 | 75.2 | 37.1 |
| EfficientNet-M-21 | 97.8 | 88.3 | 96.4 | 84.7 | 90.1 | 64.6 | 77.6 | 40.8 |
| EfficientNet-L-21 | 96.6 | 89.1 | 96.6 | 84.1 | 91.2 | 67.0 | 78.1 | 43.1 |
| ViT-T/16 [23] | 64.0 | 43.8 | 69.1 | 40.5 | 54.2 | 25.7 | 37.3 | 11.8 |
| ViT-S/16 | 93.8 | 81.0 | 92.8 | 73.6 | 82.6 | 51.7 | 65.1 | 29.2 |
| ViT-B/16 | 94.4 | 85.4 | 95.3 | 76.3 | 85.2 | 59.3 | 66.7 | 32.7 |
| ViT-L/16 | 98.3 | 94.2 | 97.5 | 90.4 | 96.4 | 79.4 | 83.9 | 51.8 |
| MAE [24] | 94.4 | 85.4 | 95.3 | 80.7 | 88.1 | 64.3 | 77.3 | 37.3 |
| MoCo-V3 [25] | 91.6 | 82.5 | 91.6 | 71.1 | 84.5 | 53.3 | 69.4 | 29.4 |
| SWSL-ResNet50 [12] | 96.1 | 92.0 | 97.4 | 87.8 | 95.1 | 76.3 | 81.7 | 51.2 |
| SWSL-ResNext101-32x16d | 97.2 | 92.7 | 97.9 | 93.6 | 95.9 | 86.3 | 86.7 | 61.0 |

Table 2: Model accuracy per recognition time subset. Models are named to include architecture, training objective, and training dataset where appropriate. ResNet-X-Y% indicates a ResNet with depth X and trained on a random Y% subset of the ImageNet-1k dataset. Model names ending in 21k were pretrained on ImageNet-21k. All other models with the exception of SWSL and CLIP models were pre-trained on the full ImageNet-1k dataset.

## A.8 Finetuned Models

Here we list details regarding training/finetuning procedures for the model results reported in the paper.

### A.8.1 Model training procedure

Pretrained models weights were instantiated using publicly available model checkpoints, either using torchvision or found on the model's source repository. The models—with the exception of CLIP—were then finetuned using subsets of the ImageNet training and validation sets containing only the 50 classes we chose to use in the psychophysics experiments. The models were finetuned for 90 epochs with an SGD optimizer and initial learning rate of 0.1 with momentum value of 0.9 and weight decay coefficient of 0.0001. The learning rate decayed by a factor of 2 every 9 epochs.

| Model | ImageNet | ObjectNet |
|---|---|---|
| ResNet-18 | 79.3 | 47.1 |
| ResNet-18-80% | 74.5 | 44.1 |
| ResNet-18-60% | 74.3 | 42.8 |
| ResNet-18-40% | 70.5 | 39.3 |
| ResNet-18-20% | 63.5 | 31.0 |
| ResNet-50 | 86.6 | 59.3 |
| ResNet-50-80% | 81.7 | 52.2 |
| ResNet-50-60% | 79.6 | 50.5 |
| ResNet-50-40% | 76.3 | 45.1 |
| ResNet-50-20% | 70.4 | 36.9 |
| ResNet-101 | 88.8 | 62.5 |
| ResNet-101-80% | 84.0 | 55.6 |
| ResNet-101-60% | 80.5 | 51.4 |
| ResNet-101-40% | 78.0 | 44.9 |
| ResNet-101-20% | 72.2 | 37.2 |
| ResNet-152 | 89.7 | 64.4 |
| ResNet-152-80% | 83.7 | 54.7 |
| ResNet-152-60% | 81.1 | 51.7 |
| ResNet-152-40% | 77.5 | 46.5 |
| ResNet-152-20% | 71.8 | 37.9 |
| CORNet-S | 80.2 | 51.6 |
| VOneNet-Resnet50 | 83.4 | 53.8 |
| VOneNet-CORNet-S | 80.5 | 47.4 |
| VGG-19 | 80.1 | 49.4 |
| Noisy Student (EfficientNet-L2) | 85.7 | 50.3 |
| DenseNet-121 | 83.7 | 55.2 |
| MSDNet Classifier 0 | 61.7 | 28.9 |
| MSDNet Classifier 1 | 74.0 | 40.9 |
| MSDNet Classifier 2 | 78.3 | 48.3 |
| MSDNet Classifier 3 | 78.3 | 50.6 |
| MSDNet Classifier 4 | 79.4 | 51.0 |
| SimCLR ResNet50 | 75.1 | 45.8 |
| SimCLR ResNet101 | 81.5 | 54.4 |
| SimCLR ResNet152 | 83.4 | 55.7 |
| CLIP-ViT-B/32 | 79.1 | 64.0 |
| CLIP-ViT-B/16 | 84.0 | 74.1 |
| CLIP-ViT-L/14 | 91.8 | 86.0 |
| CLIP-ViT-L/14@336px | 91.6 | 86.7 |
| CLIP-ResNet-50 | 69.8 | 56.5 |
| CLIP-ResNet-101 | 74.5 | 61.0 |
| CLIP-ResNet-50x4 | 76.3 | 64.9 |
| CLIP-ResNet-50x16 | 79.6 | 72.9 |
| CLIP-ResNet-50x64 | 86.6 | 80.3 |
| EfficientNet-S | 82.1 | 47.9 |
| EfficientNet-M | 80.6 | 46.2 |
| EfficientNet-L | 85.5 | 52.2 |
| EfficientNet-S-21 | 89.6 | 65.2 |
| EfficientNet-M-21 | 90.9 | 67.7 |
| EfficientNet-L-21 | 91.4 | 68.8 |
| ViT-T/16 | 58.1 | 28.9 |
| ViT-S/16 | 84.4 | 56.1 |
| ViT-B/16 | 86.6 | 60.7 |
| ViT-L/16 | 94.6 | 77.6 |
| MoCo-V3 | 85.1 | 55.9 |
| SWSL-ResNext101-32x16d | 95.0 | 83.2 |
| SWSL-ResNet50 | 93.5 | 75.3 |
| MAE-ViT-B/16 | 89.6 | 65.0 |

Table 3: Model accuracy on the ImageNet and ObjectNet subsets of our 4,771 images.

### A.8.2 Model Performance

We evaluate our finetuned models on the same cropped images used in our psychophysics experiments. See table 3 for model accuracy reports on the image difficulty reported in the paper and table 2 for model performance on the full ImageNet and ObjectNet subsets of the experiment images.

### A.9 Metric calculation procedure

In this section, we go through the details in computing c-score, prediction depth, and adversarial robustness for our experiment images.

#### A.9.1 C-score

C-score [6] identifies individual image difficulty by characterizing the expected accuracy or a held-out image given training sets of varying size sampled from the data distribution. In particular, c-score is the frequency of classifying an example correctly when it is omitted from the training set. However, computing c-score for each image by brute force is computationally infeasible since we must train a separate model for each image. Instead, we computed the learning speed proxy as recommended by the authors. Learning speed measures the epoch at which an image is correctly classified by a model. Intuitively, a training example that is consistent with the training set should be learned quickly because the gradient step for all consistent examples should be similar. The authors found high Spearman rank correlation between c-score and cumulative learning speed based proxies.

We trained a ResNet-50 [13] from scratch on ImageNet1k [9] for 90 epochs with an SGD optimizer and initial learning rate of 0.1 with momentum value of 0.9 and weight decay coefficient of 0.0001. The learning rate decayed by a factor of 2 every 9 epochs and the batch size was 256. The standard ImageNet transforms were applied to all images, and the network was initialized randomly. We then evaluated our experiment images at each epoch and used the average of correct predictions as an estimated c-score for each image. **??** shows the average c-scores for ImageNet and ObjectNet experiment images split by whether the ResNet-50 correctly predicted the image. C-score serves as an efficient predictor for human recognition difficulty only for images classified by the model in both ImageNet and ObjectNet. C-scores for images misclassified by the model do not reveal information about the human recognition difficulty and remain consistently low across all difficulty subsets.

#### A.9.2 Prediction depth

Prediction depth [7] represents the number of hidden layers after which the network's final prediction is already determined. The authors showed that prediction depth is larger for examples that visually appear to be more difficult and is consistent between architectures and random seeds.

We trained a linear decoder at the end of each block of a ResNet-50 on the 50 experiment classes using the ImageNet training and validation set. We used the same ResNet-50 used to calculate c-scores to ensure consistency of our results. There are 16 convolutional layers in a ResNet-50; and each linear decoder follows a convolution layer and consists of a pooling layer, flatten layer, and fully-connected layer. We use the same hyperparameters as appendix A.9.1 and only updated the weights of the linear decoder.

A prediction is defined to be made at depth $L = l$ if the linear classifier after layer $L = l - 1$ is different from the network's final prediction, but the classification of the linear decoder after every layer $L \geq l$ are equal to the final classification of the network. Images classified by all decoders are said to be predicted at layer 0. Note that prediction depth is independent of whether the final prediction is correct or not. It measures the layer at which an image's prediction converges.

Figure 10 shows the correlation between c-score and image viewing time difficulty.

#### A.9.3 Adversarial robustness

We measured an image's distance to the decision boundary of a network using fast gradient sign method (FGSM) [8]. FGSM creates an modified example that maximizes the loss using the gradients of loss with respect to the input image:

$$mod_x = x + \epsilon \cdot sign(\nabla_x J(\theta, x, y))$$

where $adv_x$ is the modified image, $x$ is the original image, $y$ is the original input label, $\epsilon$ is a multiplier adjusted accordingly to control the size of modification step, $\theta$ is the model parameters, and $J$ is the

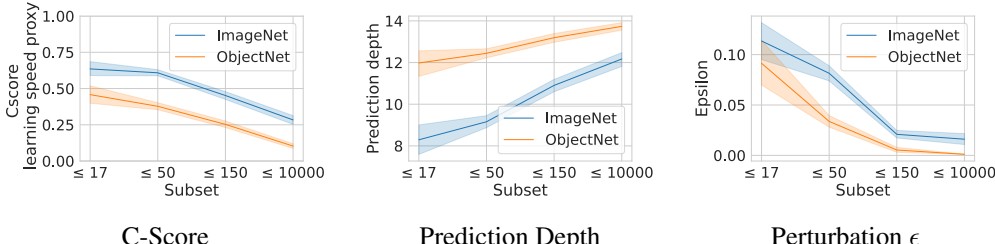

|   C-Score   |   Prediction Depth   |   Perturbation $\epsilon$   |

Figure 10: The correlation between three metrics and image difficulty. On the horizontal axis images subset by the minimum amount of viewing time, in ms, required for the majority of participants before they are recognized. All three metrics are correlated with difficulty. Hard images are learned later in model training, predicted by later layers, and need much smaller perturbations to attack.

loss function. Note that gradients are taken with respect to the input image, and model parameters remain constant.

For an image classified by a model, we define its distance to the closest decision boundary of the model as the minimum $\epsilon$ needed for the model to misclassify the modified image. On the other hand, for an image misclassified by a model, we define its distance to the closest decision boundary of the model as the minimum $\epsilon$ needed for the model to classify the modified image.

We used the same ResNet-50 used to calculate c-scores to ensure consistency of our results. We finetuned the ResNet-50 on the 50 experiment classes using the ImageNet training and validation set. We used the same hyperparameters as appendix A.9.1 and only updated the weights of the final pooling, flatten, and fully-connected layer. We used this finetuned ResNet-50 as the backbone for adversarial perturbation and correction.

While perturbing each classified image, we searched for the smallest $\epsilon$, from 0 to 0.02 incrementing by 1.25e-5 and from 0.02 to 2.5 incrementing by 0.005, that would result in a misclassification. We only applied only one gradient step when perturbing. While correcting each misclassified image, we searched for the smallest $\epsilon$, from 0 to 0.001 incrementing by 1.25e-6 and from 0.001 to 0.05 incrementing by 1.25e-5. We applied two gradient steps when correcting because correction requires finer and more steps.

Note that the search range depends on the backbone model and the dataset. One must choose them through manual trial-and-errors to yield interesting and significant results. Recall that after removing images that were incorrectly annotated, incorrectly cropped, etc **??**, we reduced to 4,771 images from the original 5,000. Of these, 3,296 and 1,475 images were classified and misclassified by the finetuned ResNet-50 respectively. We were not able to find an $\epsilon$ for every image while perturbing and correcting in the corresponding search range. We omitted these images in our analysis. We were able to successfully perturb 2,815 out of 3,296 classified images and correct 1,114 out of 1,475 misclassified images.

We hypothesized that difficult images that are classified and misclassified would be closer and further from the decision boundary respectively. fig. 10 confirms the prior hypothesis.

# B Compute usage

Roughly four weeks of compute time on two machines with eight TITAN RTX were used to generate the results presented here, largely in computing c-score, prediction depth, adversarial robustness, and finetuning models.

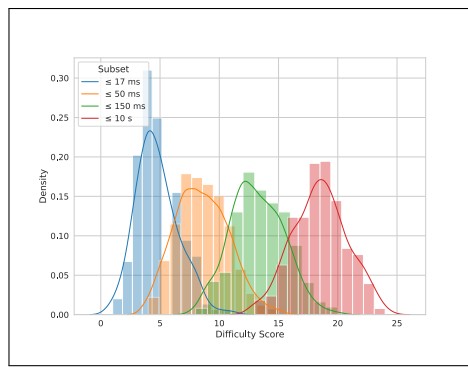 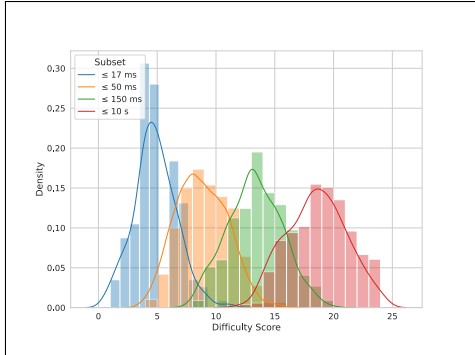

Figure 11: Distribution of difficulty score for each MRT subsets in ImageNet (left) and ObjectNet (right).

## C    Constructing a metric for image difficulty

We propose two metrics:

1. Difficulty score which provides an exact ranking from most difficult to recognize to least difficult to recognize based on each response

2. four minimum recognition time (MRT) subsets that quantify the minimum amount of time required for the majority of participants to reliably recognize an image.

Difficulty score is a value from 0 to 28 that represents the number of incorrect predictions given by participants in our experiment across all timings for a particular image. Each image in our experiment was seen an equal number of times per timing and and only rarely were images that were recognizable at shorter timings also recognizable at longer timings. This results in a low difficulty score indicating that an image is easy to recognize and a high difficulty score indicating that an image is hard to recognize. These scores correlate well with the MRT difficulty subsets as shown in fig. 11. Difficulty score varies significantly by object class as well (see fig. 12).

### C.1    Difficulty score distribution by object class

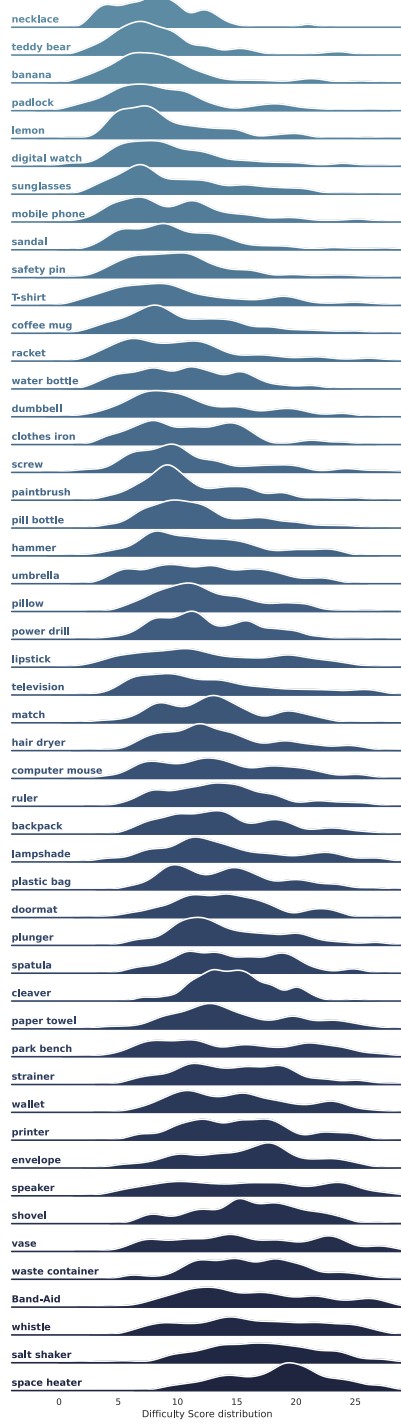

Figure 12: Difficulty distribution by object class sorted in order of increasing mean

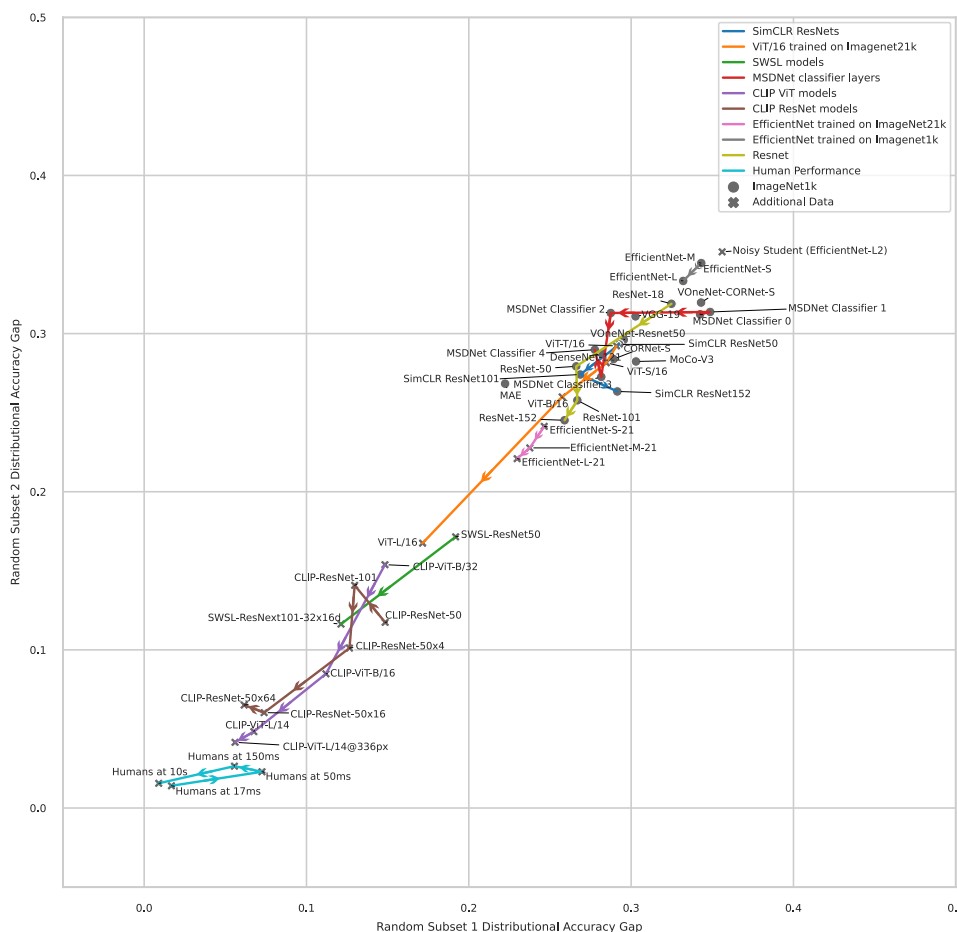

Figure 13: Robustness gap for our finetuned models on two randomly sampled subsets of our experiment data, balanced between ImageNet and ObjectNet. Lines connect model families with arrows pointing in direction of increasing model capacity. Compare with fig. 5.

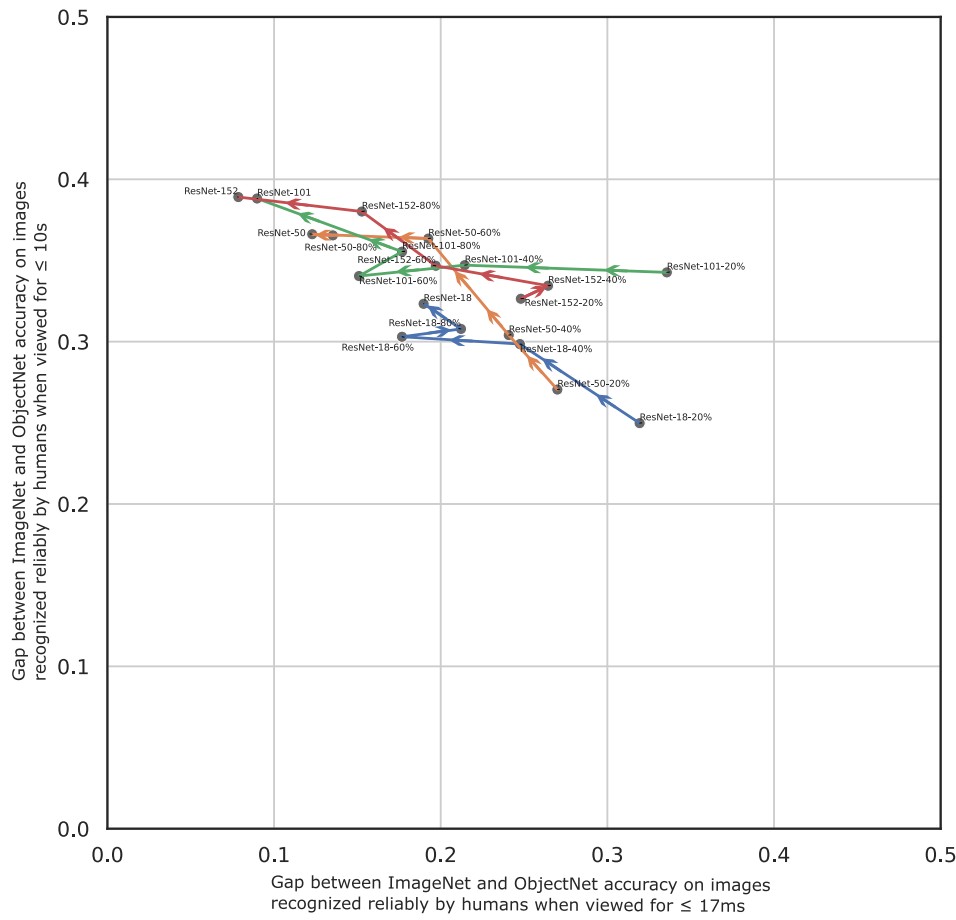

Figure 14: Robustness gap for our finetuned ResNets trained on varying percentages of the ImageNet training set. Lines connect the same architectures with arrows pointing in direction of increasing dataset percentage. Compare with fig. 5.

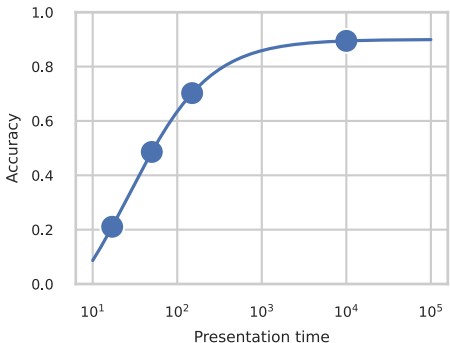

Figure 15: Human accuracy vs Image presentation time from Mechanical Turk results. Time is log-scale with a sigmoid fit

