# OpenReview forum: "Workshop version: How hard are computer vision datasets? Calibrating dataset difficulty to viewing time"
_NeurIPS.cc/2022/Workshop/SVRHM — SVRHM Poster_

### Official Review · Reviewer_LzPw · 2022-10-13
**Interesting idea to quantify image difficulty via human-looking time -- not implemented carefully enough**

**Rating:** 4
**Confidence:** 3

**Review:**

This paper introduces a novel way of quantifying the recognition difficulty of images in computer vision datasets. The metric is based on the presentation time required for humans to recognize an object depicted on a given image. The underlying rationale is that if humans take a long time to recognize an object, the image is difficult and vice versa.

The authors find that in ImageNet and ObjectNet, hard-to-recognize images are underrepresented. When looking at model performance as a function of image difficulty, the results seem to indicate that even though larger models show increased overall accuracy, this gain is mostly due to gains on easy, but not on hard images.

I like the idea of quantifying image difficulty with minimal human-looking time. Further, I think the resulting data could be important to achieve a better understanding of the similarities and differences (which are potentially masked by coarse-grained metrics such as overall classification accuracy) between biological and artificial object recognition systems. However, unfortunately, the presented paper is written and structured poorly, entails unclear and confusing plots, and has some scientific flaws. In my opinion, the paper needs a major revision. Thus, I cannot recommend accepting this paper without a potential review of the revised version. In the following, I try to point out a sample of issues that need to be addressed.

Major scientific concerns:

- It is assumed that, depending on the stimulus presentation duration and the time limit to give a response, different brain mechanisms are involved in object recognition. Core object recognition, for example, refers to the ability to rapidly recognize an object (<300ms from image onset to classification decision) and is thought to be mainly realized by feedforward mechanisms in the brain. However, increasing stimulus presentation time up to 10sec will measure something quite different as it also captures recurrent processes in the brain. This means that, depending on stimulus presentation duration, the presented experiment is measuring different abilities. I cannot find any details regarding the response time limit or reaction times. This absolutely must be mentioned. Showing a stimulus for 17ms and having unlimited time to respond is completely different than showing the same stimuli for the same duration but constraining the time to respond (consider adding response time to Figure 2). Furthermore, models like ResNets only feature feed-forward connections, so they are assumed to be models for core object recognition, but not per se of recognition processes that feature recurrent connections. Thus, be careful what you are comparing and make sure that you are not comparing apples with pears and that model-human comparison is “fair”.

- In my opinion, conducting psychophysical studies that require stimulus presentation with millisecond precision should not be conducted on AMT (especially if stimulus presentation duration determines what is measured – see point above). You wrote that true stimulus presentation times might have varied to a large extent in the ATM experiment and could have been up to **twice as long** as compared to the lab experiment (which is btw conflicting with the statement in the appendix in which you write that presentations duration might have been slightly longer; line 321). I.e., you don’t have any control over the true stimulus presentation time on AMT. This being said, I was surprised that you focused merely on the AMT experiment (line: 55). I would suggest collecting more trials from observers in a controlled lab setting.

Further points to consider:

- The plots need to be revised on many different levels. First, they are way too tiny – e.g., for Figure 6 I had to zoom in 435% to be able to see something. Second, the caption should entail all relevant information. E.g., there is no information on what the error bars represent in Figure 4. Further, it is recommended to describe all subplots of a figure separately. For example, I had a tough time understanding what is going on in Figure 4c, as it is not mentioned in the caption. In fact, the first sentence of the caption is not even true for all subplots – whereas 4a and 4b do show “Accuracy as a function of presentation time” this is not true for 4c which rather shows accuracy as a function of image difficulty for different presentation times. Further, it is misleading that the tick labels in Figures 4c and 5 are given in milliseconds, even though they refer to something different than in Figures 4a and b. I figured out that in Figures 4c and 5 they refer to a subset of images which are recognized by more than 50% of human observers within a certain presentation time. To minimize confusion, you could for example label these subsets as “super easy, easy, hard, and super hard”. Also, be careful with the color coding. In Figure 4 you use the same colors (blue and orange) in different subplots for different things (dataset vs. presentation time).

- You mention that your contribution is a novel dataset (e.g., Line: 15, 35). However, there is no link or online resource provided.

- The paper needs to be structured better. E.g., why is “Related Work” discussed at the very end? This struck me as counterintuitive.

---

### Official Review · Reviewer_1bQJ · 2022-10-13
**A well written and carried out experiment using a human difficulty measures to better understand CV datasets and to further improve and evaluate current CV approaches.**

**Rating:** 7
**Confidence:** 4

**Review:**

The paper "How hard are computer vision datasets" is well written, clear and understandable. The authors propose to use image difficulty of humans as a additional metric to help evaluate models in the future.

quality
The work described in the paper is well carried out and of good quality. The acquisition and design of the human difficulty measurement experiment seems to be adequately designed and meets the standards of the field. The selection of the presentation times could be discussed but does not harm the results or implications. The number and selection of images from the different datasets seems to be reasonable and for the purpose of this study adequate. The same is true for the selection of models and datasets. The in silico experiments seem to be adequate and of good quality. The figures are clear (although the last one is way to small) and support the story and results of the paper.

Clarity:
The paper is clearly written and easy to understand and follow. The same is true for the run experiments and the analysis visualized in the plots.


Originality:
Using a human difficulty measure to compare it with neural networks or visualize datasets does not seem like a completely novel idea. But the way the paper is written and with the additional proposed metrics that can be used to computationally get a proxy of the image difficulty make it interesting for the field and for the workshop.


Significance:
I think this paper has its clear target audience and could help the field to move towards not only focusing on accuracy as a metric to evaluate and further develop models but also to keep image difficulty perceived by humans into account.

---

### Official Review · Reviewer_JbbB · 2022-10-14
**Useful resource, medium innovative**

**Rating:** 7
**Confidence:** 4

**Review:**

Some of the in this paper can also be found, although in a less extensive form, in Kohitij Kar et al., 2019 Nature Neuroscience.
This paper is however much more focussed on the distribution of easy vs difficult images. The paper also introduces us to a substantial resource of images that have been rated as easy and difficult. The combination makes this paper a fine addition to current literature.

---

### Official Review · Reviewer_8vXU · 2022-10-14
**Clear contribution, unclear significance**

**Rating:** 6
**Confidence:** 3

**Review:**

# Summary
The authors collect a time-based behavioral measure in order to quantify human-perceived dataset difficulty. The work yielded a) a set of high-quality behavioral data that will be greatly appreciated by other researchers and b) the insight that two popular datasets (ImageNet and ObjectNet) exhibit a time-based difficulty distribution that is skewed towards easy.

# Pros:
* The manuscript makes a genuine contribution to the field in the form of a new dataset and analysis insights.
* The research objective and results are clearly described.
* The experimental design neatly circumvents issues of speed-accuracy trade-off.

# Cons:
## Major:
My primary concern is that the significance of the work is weakly communicated. The reader is given insufficient guidance on how to situate the results amongst related work. While the authors identify four categories of existing efforts, the overall narrative of the manuscript implies a mostly empty space in regards to quantifying (human-percieved) dataset difficulty. I realize that the extremely short page limit requires dropping all but the highest priority items. But on the flip side, omitting relevant work undercuts a reader's ability to assess originality. For example, how does the authors` measure complement or exceed other measures such as human confusion matrices (such as those used in Brain Score, which the authors cite)? Narrowing in on time-based measures; response times have a long history in cognitive science, from notional ideas to sophisticated computational models---it is surprising that no connection is made to this work.

## Minor:
* It seems a large logical step between discovering a skewed time-based difficulty distribution and the belief that rebalancing the training set to focus more on hard images will result in better "in the wild" performance. While I think it sounds worthwhile, the authors provide little intuition why this will pay off.
* While the macro organization was clear, some sentences were highly unusual (lines 69-70 "In-lab, even for the hardest images, humans have nearly perfect performance.") and difficult to parse (line 68 "Note that this understates human performance as it shows the Mechanical Turk results.").
* What was the motivation for the chosen time intervals?
* Figure 6 text is small.
* line 110 "Doing so with a model is free, ..." Cheaper perhaps, but compute time is not free.

I want to reiterate that the work is undeniably a contribution, but in its current form, it is left to the reader to determine the significance of the work.